# Surveillance of Avian Metapneumovirus in Non-Vaccinated Chickens and Co-Infection with Avian Pathogenic *Escherichia coli*

**DOI:** 10.3390/microorganisms12010056

**Published:** 2023-12-28

**Authors:** Gleidson Biasi Carvalho Salles, Giulia Von Tönnemann Pilati, Beatriz Pereira Savi, Eduardo Correa Muniz, Mariane Dahmer, Josias Rodrigo Vogt, Antonio José de Lima Neto, Gislaine Fongaro

**Affiliations:** 1Laboratory of Applied Virology, Department of Microbiology, Immunology and Parasitology, Universidade Federal de Santa Catarina, Florianópolis 88040-900, SC, Brazil; giuliavpilati@gmail.com (G.V.T.P.); beasavis2@gmail.com (B.P.S.); marianedahmer@gmail.com (M.D.); 2Zoetis Industry of Veterinary Products LTDA, São Paulo 04709-11, SP, Brazil; eduardo.muniz@zoetis.com (E.C.M.); josias.vogt@zoetis.com (J.R.V.); antoniovs@gmail.com (A.J.d.L.N.)

**Keywords:** clinical signs, slaughter convictions, virus–bacteria co-infection, colibacillosis

## Abstract

Brazil is the second largest producer of broiler chicken in the world, and the surveillance of avian pathogens is of great importance for the global economy and nutrition. Avian metapneumovirus (aMPV) infection results in high rates of animal carcass losses due to aerosacculitis and these impacts can be worsened through co-infection with pathogenic bacteria, particularly *Escherichia coli* (APEC). The present study evaluated the seroprevalence of the main aMPV subtypes in unvaccinated broiler chickens from poultry farms in Brazil, as well as the clinical effects of co-infection with APEC. Blood samples, respiratory swabs, femurs, liver, and spleen of post-mortem broiler chickens were collected from 100 poultry production batches, totaling 1000 samples. The selection of the production batch was based on the history of systemic and respiratory clinical signs. The results indicated that 20% of the lots showed serological evidence of the presence of aMPV, with two lots being positive for aMPV-B. A total of 45% of batches demonstrated co-infection between aMPV and APEC. The results point to the need for viral surveillance, targeted vaccination, and vaccination programs, which could reduce clinical problems and consequently reduce the use of antibiotics to treat bacterial co-infections.

## 1. Introduction

The transmission of respiratory agents in poultry farming generates constant challenges for global activity; losses are related to the decrease in zootechnical performance and direct impacts on the quality of life of affected animals [1]. The presumptive diagnosis of these diseases is difficult to perform, as there are no pathognomonic signs for viral respiratory diseases, such as avian influenza (IA), Newcastle disease (NCD), infectious bronchitis virus (IBV) and avian metapneumovirus (aMPV) [2].

The aMPV is a virus belonging to the Pneumoviridae family, *Metapneumovirus* genus, which mainly affects the respiratory and reproductive systems of birds when infected [3]. The classification of aMPV can be carried out based on its envelope glycoproteins (G, F, and SH), the main of which is the G glycoprotein, which is responsible for binding to the host cell receptor [4]. The distinction in some amino acids present in the genetic material can alter the subtypes of aMPV; only four subtypes are described based on their antigenicity: A, B, C, and D [5]. Two intermediate subtypes have also been described [6]. Subtypes A and B are more similar to each other than subtype C, for example [5]. In Brazil, the first reported case of aMPV occurred in the mid-1990s [7], although this disease is relatively new in the country and few epidemiological studies have been developed. The aMPV has already been identified on most continents and its first description occurred in South Africa in turkeys, as TRT (Turkish Rhinotracheitis) [8]. In just a few years, it has been possible to identify aMPV in several different regions since its first appearance. A factor that can significantly contribute to this spread, in addition to migratory birds, is the intercontinental movement of people [9].

The aMPV Infection in turkeys and chickens continues to be a serious problem for producers worldwide. Subtypes A and B are the most prevalent and are responsible for causing the greatest production losses, particularly when associated with the swollen head syndrome, which manifests as edema of the periorbital and infraorbital sinuses, along with mucus production and nasal secretion [1,10]. The problems are not confined to the respiratory system; the two main subtypes can also affect egg production and quality due to their preference for replicating in tissues of the respiratory and genitourinary tracts [11]. Although vaccines are available to prevent aMPV-A and aMPV-B in Brazil, the immunoprophylaxis strategy to prevent this agent is not commonly employed, especially in broiler chickens.

Upper respiratory tract infections caused by aMPV can be isolated or, in many cases, associated with bacteria such as *Escherichia coli* [1]. Coinfection-related viral damage and persistence may be altered compared to viral monoinfections [12]. Infections caused by avian pathogenic *Escherichia coli* (APEC) can be primary or secondary [13,14]. The development of secondary infections by APECs is conditioned by predisposing factors that can disturb the host’s organic balance, such as compromised integrity of the skin or mucous membranes, poor hygiene practices, influence of immunosuppressive factors, inadequate ventilation, and the presence of viral diseases [13,14,15].

In this context, a primary viral infection of the airways can lead to a secondary bacterial infection. The increase in bacterial binding factors induced by the virus favors the clinical manifestation caused by *Escherichia coli* [16], and the damage generated by viral replication in mucociliary tissues favors bacterial maintenance in the respiratory tract [17,18].

Regarding the economy and poultry production, Brazil is the second largest producer and largest exporter of chicken meat in the world, being evidenced in global health and nutrition. In relation to the slaughter of broiler chickens, the states of Paraná, Santa Catarina and Rio Grande do Sul, São Paulo, Goiás, and Minas Gerais stand out (together they represent 88.33% of the total birds slaughtered in Brazil and exported) [19]. The state of Ceará allocates its production for Brazilian domestic consumption [19]. This high percentage of birds housed in a geographic region can pose health risks to the animals’ health, mainly through the transmission of respiratory infectious agents [20]. An epidemiological study conducted between 2004 and 2008 in Brazil, involving 228 samples from broilers, broiler breeders, and turkeys, revealed a prevalence of 6.57% for aMPV-A and 10.08% for aMPV-B [7].

In view of the above, the present study aimed to evaluate the seroprevalence of aMPV in unvaccinated broiler chickens, perform molecular detection by RT-PCR, and identify the subtypes present in Brazil, in addition to evaluating the impacts caused in batches of broiler chickens that presented co-infection between aMPV and APEC.

## 2. Materials and Methods

### 2.1. Sample Collections

A total of 100 batches of broiler chickens were evaluated (*Gallus gallus domesticus*), distributed throughout Brazil. Collections were carried out between August and December 2021. The definition of the states where the samples were collected was with respect to the proportionality of broiler chicken production; ten chickens were sampled per batch, totaling one thousand chickens, coming from the South Region (states of Paraná (*n* = 30 batches), Santa Catarina (*n* = 15 batches), Rio Grande do Sul (*n* = 15 batches), Southeast Region (states of São Paulo (*n* = 10 batches), and Minas Gerais (*n* = 10 batches)) and Northeast Region (state of Ceará (*n* = 20 batches)), which represent 80% of chicken meat production in Brazil [19]. Figure 1 shows the regions and collection areas.

The batches were selected based on the history of respiratory problems and animals that presented some respiratory disorder, such as sneezing, rales, snoring, nasal secretions and swollen head, and suspected colibacillosis, based on the clinical assessment by the responsible veterinarians at each farm. Furthermore, the birds were not vaccinated for aMPV.

The samples were collected from chicken carcasses, aged between 13 and 32 days. The chickens were necropsied in the field for routine health inspection. For each batch, ten nasotracheal swabs, three livers, three spleens, and three femurs were collected. Liver and spleen were evaluated, taking into account the presence of macroscopic lesions). Samples were stored at a temperature of 2 °C to 8 °C. Respiratory swabs were eluted in saline buffer for the purpose of diagnosing aMPV, while bone marrow was collected from the femurs to detect APEC.

All biological samples evaluated here were donated by farms that carry out routine inspections, eliminating the need for an ethics committee, as they are leftover biological samples collected by routine health surveillance services—Consultation with the Ethics Committee on the Use of Animals (no 4434190521/Federal University of Santa Catarina).

For the purpose of serological evaluation for aMPV, blood samples were collected 15 to 21 days after the first collection. Collection was carried out in pools of 20 animals and the sera were stored individually by batches.

### 2.2. Clinical Signs in Batches

To survey the clinical signs of the sampled batches, anamnesis was carried out and the individual sanitary control sheets of the batches were evaluated, where information such as medications used, clinical signs, average weight, and feed consumption were recorded.

### 2.3. Serological Detection of aMPV

To detect antibodies against aMPV, the ELISA (Enzyme Linked Immuno Sorbent Assay) method was used, using the BioChek commercial kit (Reeuwijk, The Netherlands), following the manufacturer’s instructions. The Freeze-Dried Reference Serum CR300 (BioChek) commercial kit (Reeuwijk, The Netherlands) was used as positive control. The results were analyzed according to an optical density (OD) using the BioChek ii software (version 2015) with sample/positive ratios (SP) > 0.5 (titer ≥ 0.501), indicating the average titer of the 20 birds evaluated per batch against possible natural exposure to aMPV, since they are not birds vaccinated against the pathogenic agent.

### 2.4. Molecular Detection of aMPV by RT-PCR

Total RNA from the samples was extracted using the RNeasy^®^ Mini kit (QIAGEN, Hilden, Germany) following the manufacturer’s instructions. The M-MLV Reverse Transcriptase kit (Promega, Madison, WI, USA) was used to perform reverse transcription, following the manufacturer’s instructions. For the polymerase chain reaction (PCR), the G protein gene was used for the detection of subtypes A and B (Table 1), using the following reagents and concentrations: 2 mM magnesium chloride, 0.25 mM deoxyribonucleotide phosphates, 0.3 μM of each primer, 1 U of Taq DNA polymerase GoTaq^®^ DNA Polymerase (Promega, Madison, WI, USA), 1× Green GoTaq^®^ Reaction Buffer (Promega, Madison, WI, USA), 3 μL of sample and sterile ultrapure water, to make 25 μL. The reactions were carried out in a thermocycler, using the following cycling parameters: 94 °C for 2 min; 35 cycles of 94 °C for 30 s, 63 °C for 30 s, 68 °C for 3 min; and a final cycle of 72 °C for 10 min [21].

The samples were subjected to horizontal electrophoresis in 1% agarose gel, using GelRed (MilliporeSigma™, Burlington, MA, USA) as a DNA intercalating agent. Amplicon sizes were determined by comparison with the low-molecular-weight (LMW) marker.

### 2.5. Assessment of APEC Co-Infection

For the isolation of *Escherichia coli*, femur swabs were inoculated on MacConkey agar and incubated at 37 °C for 24 h. Typical *Escherichia coli* colonies were confirmed as APEC using qualitative PCR, as described by [22,23] using the genes *iroN*, *ompT*, *hlyF*, *iss*, and *iutA* as the predictors of APEC virulence (Table 2).

## 3. Results

### 3.1. Clinical Signs and Lesions in Batches

A total of 43 batches showed no clinical signs (attribute “0”), 29 batches showed only one clinical sign (attribute “+”), and 28 batches showed the presence of more than two clinical signs (attribute “++”). The ranking of the batches is shown in Table 3, evaluated according to the clinical signs obtained from the batch health control sheets, containing information on the clinical signs observed and medications used.

During sample collection, it was observed that 71% of the batches showed clinical respiratory signs, including rales, sneezing, nasal discharge, enlargement of the infraorbital sinus, and swollen head. Among the batches from the southern states (Santa Catarina and Paraná), 13.3% used antibiotics during the birds’ housing phase; among the drugs used were ciprofloxacin, sulfachlorpyridazine+trimethopim, and florfenicol. In the Southeast region (São Paulo and Minas Gerais), only one batch (5%) of the batches showed clinical signs, and this batch was medicated with ciprofloxacin on the day of collection. The clinical signs observed were different between the batches, when the batch control sheets were checked.

The states of Rio Grande do Sul, Paraná, and Santa Catarina, which make up the southern region of Brazil, stand out, with 83.3% of the batches showing various respiratory clinical signs. Additionally, 36.6% of these batches used antibiotic therapy at some point in the production cycle. On the other hand, in the states of São Paulo and Minas Gerais, representing the Southeast region, only 5% of the batches showed clinical signs of respiratory diseases, coinciding with the use of antibiotics in one of these batches. In the state of Ceará, representing the Northeast region, all batches collected exhibited clinical signs at the time of sampling, and 20% of them were under drug treatment. This diversity of scenarios highlights the variation in the prevalence of symptoms and the use of antibiotics in different regions of the country.

In the macroscopic evaluations of the animals’ organs, a total of 90% (90/100) of the batches exhibited alterations in the spleen, liver, or air sacs, such as splenomegaly, hepatomegaly, changes in coloration and opacity, and vascularization of the air sacs. These alterations were observed in 100% (15/15) of the bird batches from Rio Grande do Sul, Minas Gerais (10/10), São Paulo (10/10), and Ceará (20/20), in 93% (14/15) from Santa Catarina, and 70% (21/30) from the state of Paraná. This demonstrates the impact of coinfections, particularly involving viruses and bacteria.

### 3.2. Seropositivity and Molecular Detection of aMPV

As a result, 20% of the samples showed the presence of antibodies against aMPV. The positive samples were concentrated in the southern region of Brazil, as 70% came from Paraná and the remaining 30% from Santa Catarina. In the State of Paraná, the results indicate positivity in 14 of the 30 batches sampled, revealing a seroprevalence for aMPV of 46.6% of the batches evaluated, with an average titratable weight of 6881.4 IU. In Santa Catarina, the results indicate positivity in six of the fifteen batches sampled, with aMPV seroprevalence of 40% of the batches evaluated, with a titratable average of 780 IU (Figure 2).

Of the one-hundred batches evaluated for detection and molecular typing of aMPV by RT-qPCR, two batches were positive for the aMPV-B subtype, while no batches were positive for aMPV-A. Both positive samples came from the state of Paraná, which had positive serology with serological titers of 3.909 and 4.821 IU.

### 3.3. Escherichia coli Detection and APEC Molecular Confirmation

A collection of 63 characteristics of *E. coli* isolates was acquired from the femurs. Employing qualitative PCR, it was determined that, out of the 63 *E. coli* isolates, 58 (92%) manifested three to five of the genes recognized as minimum virulence indicators for APEC strains.

### 3.4. Co-Infection between aMPV and APEC

All batches with serology compatible with aMPV were tested for the presence of APEC. Of the batches in which APEC was isolated from femurs, 20% (13/63) presented antibodies against aMPV in the ELISA assay, and in one batch, the genetic material of type B aMPV was detected.

Clinical signs in these animals were generally more severe compared to either disease alone. The batches that showed co-infection between aMPV-B and APEC came from the states of Paraná and Santa Catarina (Table 4).

## 4. Discussion

The present study demonstrated seroconversion to aMPV in batches of broiler chickens not vaccinated against aMPV in the southern region of Brazil, specifically in the states of Paraná and Santa Catarina, which are the main poultry producers in the country, occupying the first and second positions, respectively [19]. In this study, 20/100 batches demonstrated seroconversion to aMPV and 2/100 of these were characterized as aMPV-B using RT-PCR. The detection of the viral genome and isolation of aMPV represents a considerable challenge, since the virus has a relatively short period of persistence in the host and is often detected in the early stages of infection, without demonstrating characteristic clinical signs [24].

The high density of poultry farms in certain regions and the frequent non-use of vaccines to prevent aMPV allows viral spread in poultry flocks. It is also noteworthy that in the southern region of Brazil, there is intense production of turkeys, which may also contribute to the spread and maintenance of the virus in the region, considering that subtypes A and B can be found mainly in turkeys and chickens [25].

The aMPV is widely distributed worldwide [2,26,27]. In Latin America, the first report occurred in 1995 [28]; using field samples of aMPV and cells derived from chicken embryos, they identified subtype A. At the first appearance, an increase in cases of aMPV, mainly in long-lived turkeys and chickens, was observed. In 2011, ref. [29] characterized the first appearance of aMPV-B in Brazil.

Despite being present in poultry flocks and often neglected in broiler chickens, aMPV causes significant damage in poultry farming. It was revealed that aMPV, after viral infection, causes thickening of the tracheal mucosa [30]. This occurs due to congestion, edema, and infiltration of mononuclear cells in the lamina propria of the trachea, generally appearing between three and seven days after infection. Furthermore, flattening of epithelial cells and focal disciliation were observed, which may facilitate the emergence of secondary infections, worsening clinical signs.

The detection of aMPV-B in Brazilian poultry flocks may be related to the massive use of vaccines against aMPV-A for many years, which may have exerted vaccine pressure generating alternation of aMPV-B. It is worth mentioning that aMPV-A has more limited transmission, as it is via the oral–fecal route, while MPV-B is respiratory, making it more easily disseminated [24].

Although replicating subtype A and subtype B (live) vaccines are available for use in immunoprevention, both are cross-protective [31,32,33]; however, they are not used in all states in Brazil. Some regions use replicating vaccines to prevent aMPV in broiler chickens, such as the Southeast (São Paulo and Minas Gerais) and Northeast (Ceará) regions, although the batches were collected from farms that do not use them. But this practice is relatively common in these regions. This factor may explain the low circulation of aMPV in these locations, since vaccination generates selection and control pressure, reducing clinical signs and viral excretion when used, although in the state of Ceará, 100% of batches showed clinical respiratory signs, so another viral agent must be present at that time.

In the states of the southern region (Paraná, Santa Catarina, and Rio Grande do Sul), of the 20 batches in which the presence of antibodies to aMPV was detected, only two did not show clinical signs at the time of collection, which may be linked to the characteristic of infection and viral replication in early stages [34,35,36].

Regarding the use of medicines during production cycles, there is a significant concentration in the southern region of Brazil, which represented 88% of all batches medicated in Brazil. These treatments were primarily aimed at controlling opportunistic bacteria or those naturally present in birds. Isolates confirmed as APEC were obtained in 45% of the batches in which there was seroconversion to aMPV, demonstrating the condition of co-infection. This was linked to the clinical condition of the birds, which leads to production losses during the life of the flock, as well as the loss of slaughtered birds.

The identification and characterization of APEC in aMPV-positive batches in Brazilian states demonstrates the importance of this agent, regardless of its primary or secondary role, especially in batches that were medicated to reduce impacts related to co-infection with aMPV associated with APEC [37].

## 5. Conclusions

The study presents the seroprevalence of aMPV in 20% (20/100) of the batches evaluated in Brazil, with the presence of subtype B detected, with 45% (9/20) demonstrating greater clinical problems in the presence of APEC co-infection.

This study points to the need to design constant monitoring strategies aimed at combating aMPV in the poultry sector, as well as reducing viral circulation and bacterial co-infections. This, together, will certainly have a positive impact on production, with a view to protecting livestock, improving animal health, and consequently reducing the use of antimicrobials.

## Figures and Tables

**Figure 1 microorganisms-12-00056-f001:**
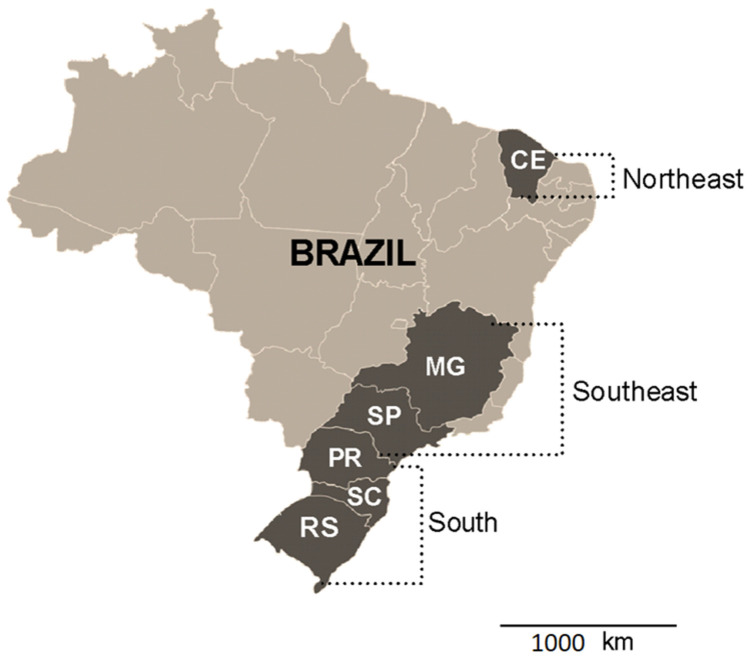
Map of Brazil, highlighting the South, Southeast, and Northeast Regions and the Brazilian states sampled in the present study, being Rio Grande do Sul (RS), Santa Catarina (SC), Paraná (PR), São Paulo (SP), Minas Gerais (MG), and Ceará (CE).

**Figure 2 microorganisms-12-00056-f002:**
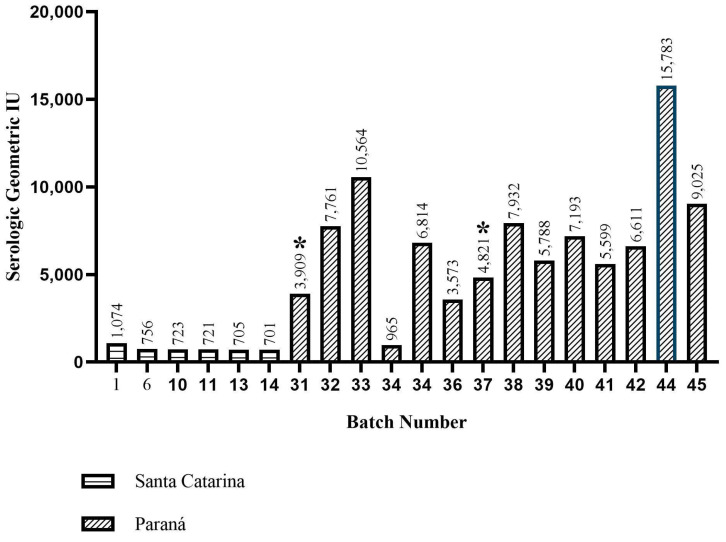
Serology for aMPV from different Brazilian states batches. (*) Batches with RT-qPCR positivity for aMPV-B.

**Table 1 microorganisms-12-00056-t001:** Primers, gene target, and size of gene fragments used in molecular detection of aMPVA A and B.

Virus	Target Gene	Primer Sequence	Amplicon Size	Ref.
aMPV/A	G protein	F 5′-GGACATCGGGAGGAGGTACA-3′ R 5′-CACTCCTCTAACACTGACTGTTCAACT-3′	116 bp	[21]
aMPV/B	G protein	F 5′-TCATCCCGGAAGCCTCCCTCACTAT-3′ R 5′-TAGCGTTTGCTGCACTGGCTTCTGATAC-3′	135 bp	[21]

**Table 2 microorganisms-12-00056-t002:** Primers, genes, and size of gene fragments used in APEC detection.

Target Gene	Primer Sequence	Amplicon Size	Reference
*iroN*	F 5′-AAGTCAAAGCAGGGGTTGCCCG-3′ R 5′-GATCGCCGACATTAAGACGCAG-3′	667 bp	[23]
*ompT*	F 5′-TCATCCCGGAAGCCTCCCTCACTACTAT-3′ R 5′-TAGCGTTTGCTGCACTGGCTTCTGATAC-3′	496 bp	[22]
*hlyF*	F 5′-GGCCACAGTCGTTTAGGGTGCTTACC-3′ R 5′-GGCGGTTTAGGCATTCCGATACTCAG-3′	450 bp	[22]
*iss*	F 5′-CAGCAACCCGAACCACTTGATG-3′ R 5′-AGCATTGCCAGAGCGGCAGAA-3′	323 bp	[23]
*iutA*	F 5′-GGCTGGACATCATGGGAACTGG-3′ R 5′-CGTCGGGAACGGGTAGAATCG-3′	302 bp	[22]

**Table 3 microorganisms-12-00056-t003:** Classification of batches according to clinical signs, injuries, medication used, and origin of samples.

Ranking Scores	Clinical Signs and Injuries	Batches	Total Batches Medicated	Medications
**0**		*n* = 29 (13, 40, 46, 48, 49, 50, 51, 52, 53, 54, 61, 62, 63, 64, 65, 66, 67, 68, 69,70, 71, 73, 74, 75, 76, 77, 78, 79, 80)	0	Unmedicated
**+**	sneezing, crackles, nasal discharge, aerosacculitis, nasal discharge,	*n* = 42 (1, 2, 3, 4, 5, 6, 7, 8, 10, 12, 16, 19, 20, 21, 27, 28, 34, 35, 37, 38, 39, 47, 55, 56, 81, 82, 84, 85, 86, 87, 88, 89, 90, 91, 92, 93, 94, 95, 97, 98, 99, 100)	3	Sultfa + Trimethoprim, Ciprofloxacin, Norfloxacin, Florfenicol,
**++**	sneezing and mucopurulent nasal discharge, sneezing and rales, rales and swelling in the periocular region, swollen head and rales, sneezing and presence of airsacculitis, sneezing, rales and nasal discharge, sneezing and tracheitis, airsacculitis and colibacillosis, septicemia, suspected colibacillosis.	*n* = 29 (9, 11, 14, 15, 17, 18, 22, 23, 24, 25, 26, 29, 30, 31, 32, 33, 36, 41, 42, 43, 44, 45, 57, 58, 59, 60, 72, 83, 96)	21	Ciprofloxacin, Sulfachlorpyridazine + Trimethopim, Norfloxacin, Florfenicol,

Classification of clinical conditions in birds based on clinical signs and lesions observed, where “0”: no clinical signs (*n* = 29), “+”: only 1 clinical sign observed (*n* = 42), and “++”: 2 or more signs observed (*n* = 29). The origin of the samples by state is identified as per the following description: 1–15 batches from the state of Santa Catarina; 16–30 batches from the state of Rio Grande do Sul; 31–60 batches from the state of Paraná; 61, 62, 65, 66, 67, 69, 75, 76, 77, 78, 80 batches from the state of Minas Gerais; 62, 63, 64, 68, 70, 71, 72, 73, 74, 79 batches from the state of São Paulo; 81–100 batches from the state of Ceará. It is also possible to check the total number of medicated batches and the medications used by the group.

**Table 4 microorganisms-12-00056-t004:** Co-infection between aMPV and APEC and classification of different levels of clinical signs/lot lesions: (0) no clinical signs, (+) only one clinical sign, and (++) 2 or more clinical signs.

Batches	Brazilian State	Serology aMPV (UI/mL)	APEC	Injury Scores by Clinical Signs
1	Santa Catarina	1074	Yes	+
6	Santa Catarina	756	Yes	+
31	Paraná	3909	Yes	++
32	Paraná	7761	Yes	++
33	Paraná	10,564	Yes	++
38	Paraná	7932	Yes	++
39	Paraná	5788	Yes	+
40	Paraná	7194	Yes	0
45	Paraná	9025	Yes	++

## Data Availability

Data are contained within the article.

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
