# Peer review of "Surveillance of Avian Metapneumovirus in Non-Vaccinated Chickens and Co-Infection with Avian Pathogenic Escherichia coli"

_microorganisms, 2023, doi:10.3390/microorganisms12010056_

Round 1

Reviewer 1 Report

Comments and Suggestions for Authors

The present study reports on the avian metapneumovirus and E. coli infections in chickens in Brazil. I believe that there is a significant scope of improvement in writing. Please see the suggestions here:

1.      English can be significantly improved in the Introduction section.

2.      L49-57: Please discuss how many subtypes have been reported? Do they vary in clinical signs of disease, virulence etc.?

3.      Escherichia coli should be written in italics at all places.

4.      L71-75: References needed.

5.      L79-82: Please provide information, if available, on previous studies in the region and the country investigating the prevalence of this virus. Also please discuss what new information the present study may add to the existing literature.

6.      L85: All scientific names, such as Gallus gallus domesticus must be italicized.

7.      L85: How do you define a batch or a lot?

8.      L93: Figure 1 is missing, only caption is provided.

9.      Also please provide the information in methods section about the age of chickens tested and time period of sampling/season etc.

10.   L100-103: I struggled with the missing information in methods section. Were the samples collected only from those chickens that had been either died or euthenized based on clinical signs of disease? There is ambiguity on the information on type(s) of samples that were collected from 1000 chickens as stated.

11.   L108-109: “femurs and 10 nasotracheal swabs were collected”. The way it is written is not clear about the sampling strategy or what was exactly done.

12.   What were the strains used as controls for ELISA?

13.   PCR results of E. coli testing are missing. Please provide gel images indicating expected amplicon sizes for various genes, as claimed in the text. Did authors attempt to perform Sanger sequencing of PCR amplicons? That will significantly improve the presented work.

14.   Also please provide gel images for virus detection. Sanger sequencing for the confirmation of subtypes would be useful.

15.   L214: Please define aMPV and aMPV-B?

16.   L216: ‘residence’ is not a right term to use in this context.

17.   Pretty much no discussion on E. coli infectivity or prevalence in chickens. Also, no justification is provided why vaccine strain aMPV-B could be detected in non-vaccinated chickens.

18.   Also please provide information on the necropsy that were conducted – any macroscopic or microscopic lesions noticed? Any histopathology done?

19. Abstract needs rewriting.

Comments on the Quality of English Language

Extensive editing of English language is required. There are serious errors in writing that must be improved. I suggest proof-reading by a native English speaker or a professional service.

Author Response

Florianópolis-SC, Brazil - 21th December  2023

Response Letter

 Manuscript ID: microorganisms-2726232

Title: Surveillance of Avian Metapneumovirus in non-vaccinated chickens and

co-infection with Avian Pathogenic Escherichia coli

Dear reviewer (Revisor 1),
Our team appreciates your time and expertise in correcting our article and improving this production.

Questions and responses:

The present study reports on the avian metapneumovirus and E. coli infections in chickens in Brazil. I believe that there is a significant scope of improvement in writing. Please see the suggestions here:

  1. English can be significantly improved in the Introduction section.

Response: Thank you for the observation. The English was corrected and improved. 

  1. L49-57: Please discuss how many subtypes have been reported? Do they vary in clinical signs of disease, virulence etc.?

Response: Thank you for the observation. 

The following paragraph was improved in the Introduction section:
(L49-58) “The aMPV infection in turkeys and chickens continues to be a serious problem for producers worldwide. Subtypes A and B are the most prevalent and are responsible for causing the greatest production losses, particularly when associated with the swollen head syndrome, which manifests as edema of the periorbital and infraorbital sinuses, along with mucus production and nasal secretion [10,1]. The problems are not confined to the respiratory system; the two main subtypes can also affect egg production and quality due to their preference for replicating in tissues of the respiratory and genitourinary tracts [11]. Although vaccines are available to prevent aMPV-A and aMPV-B in Brazil, the immunoprophylaxis strategy to prevent this agent is not commonly employed, especially in broiler chickens.”

  1. Escherichia coli should be written in italics at all places.

Response: Thank you for the observation. The name has been italicized.

  1. L71-75: References needed.

Response: Thank you for the observation. The following reference was added: Brazilian Association of Animal Protein. (2023). Annual Report. https://abpa-br.org/wp-content/uploads/2023/04/Relatorio-Anual-2023.pdf number “[19]”. 

  1. L79-82: Please provide information, if available, on previous studies in the region and the country investigating the prevalence of this virus. Also please discuss what new information the present study may add to the existing literature.

Response: Thank you for the observation. 

To clarify this point the paragraph was added:

(L: 79-82)  “An epidemiological study conducted between 2004 and 2008 in Brazil, involving 228  samples from broilers, broiler breeders, and turkeys, revealed a prevalence of 6.57% for aMPV-A and 10.08% for aMPV-B.” 

Reference: Chacón, J. L., Brandão, P. E., Buim, M., Villarreal, L., & Piantino Ferreira, A. J. (2007). Detection by reverse transcriptase-polymerase chain reaction and molecular characterization of subtype B avian metapneumovirus isolated in Brazil. Avian Pathology, 36(5), 383-387. https://doi.org/10.1080/03079450701589142

  1. L85: All scientific names, such as Gallus gallus domesticus must be italicized.

Response: Thank you for the observation. Scientific and Latin names have been corrected and placed in italics.

  1. L85: How do you define a batch or a lot?

Response: The term was standardized for “batch”.

  1. L93: Figure 1 is missing, only a caption is provided.

Response: Thank you for the observation. The image was added to the manuscript.

  1. Also please provide the information in methods section about the age of chickens tested and time period of sampling/season etc.

Response: Thank you for the observation. The two follow information were added: 

(L:98-99) “Collections were carried during August and December of 2021.”

(L: 114-115) “The samples were collected from chicken carcasses, aged between 13 and 32 days. The chickens necropsied in the field for routine health inspection.”

  1. L100-103: I struggled with the missing information in the methods section. Were the samples collected only from those chickens that had been either died or euthanized based on clinical signs of disease? There is ambiguity on the information on type(s) of samples that were collected from 1000 chickens as stated.

Response: Thank you for the suggestion. Furthermore, the birds were not vaccinated for aMPV. To clarify this point the following information was improved: 

(L: 114-119) “The samples were collected from chicken carcasses, aged between 13 and 32 days. The chickens were necropsied in the field for routine health inspection. For each batch, ten nasotracheal swabs, three livers, three spleens, and three femurs were collected. Liver and spleen were evaluated, taking into account the presence of macroscopic lesions). Samples were stored at a temperature of 2° to 8° C. Respiratory swabs were eluted in saline buffer for the purpose of diagnosing aMPV, while bone marrow was collected from the femurs to detect APEC.”

  1. L108-109: “femurs and 10 nasotracheal swabs were collected”. The way it is written is not clear about the sampling strategy or what was exactly done.

Response: Thank you for the suggestion. The paragraph was restructured:

(L: 114-119) “The samples were collected from chicken carcasses, aged between 13 and 32 days. The chickens necropsied in the field for routine health inspection.. For each batch, ten nasotracheal swabs, three livers, three spleens and three femurs were collected. Liver and spleen were evaluated, taking into account the presence of macroscopic lesions). Samples were stored at a temperature of 2° to 8° C. Respiratory swabs were eluted in saline buffer for the purpose of diagnosing aMPV, while bone marrow was collected from the femurs to detect APEC.”

  1. What were the strains used as controls for ELISA?

Response: Thanks for the observation. The following information was added:
(L: 136-137) “The Freeze Dried Reference Serum CR300 (BioChek commercial kit (Netherlands) was used as positive control”.

  1. PCR results of E. coli testing are missing. Please provide gel images indicating expected amplicon sizes for various genes, as claimed in the text. Did authors attempt to perform Sanger sequencing of PCR amplicons? That will significantly improve the presented work.

Response: Thanks for the observation. A topic covering results related to Escherichia coli has been added. 

Section: “Escherichia coli detection and APEC molecular confirmation”

(L:246-251): A collection of 63 characteristics of E. coli isolates was acquired from the femurs. Employing qualitative PCR, it was determined that, out of the 63 E. coli isolates, 58 (92%) manifested three to five of the genes recognized as minimum virulence indicators for APEC strains. Figure 3 represents one of the gels performed, testing the five genes.” 

  1. Also please provide gel images for virus detection. Sanger sequencing for the confirmation of subtypes would be useful.

We appreciate the suggestion.

Below we send the gel with all tested beaches, as well as the positive sample and positive control. If relevant, we can include supplementary materials.

Sequencing was not the objective of the study. The PCR was sufficient to confirm aMPV-A and aMPV-B. 

  1. L214: Please define aMPV and aMPV-B?

Response: The term has been correct to aMPV-B (type of virus). The information was confirmed in the manuscript.

  1. L216: ‘residence’ is not a right term to use in this context.

Response: The term was changed. (L: 244) “persistence”.  

  1. Pretty much no discussion on E. coli infectivity or prevalence in chickens. Also, no justification is provided why vaccine strain aMPV-B could be detected in non-vaccinated chickens.

Thank you for the observation.

In the two paragraphs below, these points were discussed.

“L253 - L257: The high density of poultry farms in certain regions and the frequent non-use of vaccines to prevent aMPV, allows viral spread in poultry flocks. It is also noteworthy that in the southern region of Brazil there is intense production of turkeys, which may also contribute to the spread and maintenance of the virus in the region, considering that subtypes A and B can be found mainly in turkeys and chickens [25].

L270 - L274: The detection of aMPV-B in Brazilian poultry flocks may be related to the massive use of vaccines against aMPV-A for many years, which may have exerted vaccine pressure generating alternation of aMPV-B. It is worth mentioning that aMPV-A has more limited transmission, as it is via the oral-fecal route, while MPV-B is respiratory, making it more easily disseminated [24].

  1. Also please provide information on the necropsy that were conducted – any macroscopic or microscopic lesions noticed? Any histopathology done?

Responde: Thank you for the observation.

To clarify this point, the informations were added: 

L197:L203: In the macroscopic evaluations of the animals' organs, a total of 90% (90/100) of the batches exhibited alterations in the spleen, liver, or air sacs, such as splenomegaly, hepatomegaly, changes in coloration and opacity, and vascularization of the air sacs. These alterations were observed in 100% (15/15) of the bird batches from Rio Grande do Sul, Minas Gerais (10/10), São Paulo (10/10), and Ceará (20/20), in 93% (14/15) of Santa Catarina, and 70% (21/30) of the state of Paraná. This demonstrates the impact of coinfections, particularly involving viruses and bacteria.

  1. Abstract needs rewriting.

Response: Thank you for the observation.The abstract was rewritted: 

“Brazil is the second largest producer of broiler chicken in the world, and the surveillance of avian pathogens is of great importance for the global economy and nutrition. Avian metapneumovirus (aMPV) infection results in high rates of animal carcass losses due to aerosacculitis and these impacts can be worsened through co-infection with pathogenic bacteria, particularly Escherichia coli (APEC). The present study evaluated the seroprevalence of the main aMPV subtypes in unvaccinated broiler chickens from poultry farms in Brazil, as well as the clinical effects of co-infection with APEC. Blood samples, respiratory swabs, femurs, liver and spleen of post-mortem broiler chickens were collected from 100 poultry production batches, totaling 1000 samples. The selection of the production batch was based on the history of systemic and respiratory clinical signs. The results indicated that 20% of the lots showed serological evidence of the presence of aMPV, with 2 lots being positive for aMPV-B. A total of 45% of batches demonstrated co-infection between aMPV and APEC. The results point to the need for viral surveillance, targeted vaccination, and vaccination programs, which could reduce clinical problems and consequently reduce the use of antibiotics to treat bacterial co-infections.”

Reviewer 2 Report

Comments and Suggestions for Authors

- latin names must be in italic - correct it in all manuscript!

Line 139 - Table 1 - the difference in the length of the forward and reverse primer for aMPV/A is considerable, are they listed correctly in the text?

It is necessary to add the permit number of the ethical commission, as well as the method of killing the animals used

Author Response

Dear reviewer (Revisor 2),
Our team appreciates your time and expertise in correcting our article and improving this production.

Questions and responses:

- latin names must be in italic - correct it in all manuscript!

Response: Thank you for the observation. The name has been italicized.

Line 139 - Table 1 - the difference in the length of the forward and reverse primer for aMPV/A is considerable, are they listed correctly in the text?

Response: Thank you for the observation. We checked and the primers are listed correctly

It is necessary to add the permit number of the ethical commission, as well as the method of killing the animals used

Response: Thank you for the observation. The following paragraph was added: 

(L: 114-117):  “All biological samples evaluated here were donated by farms that carry out routine inspections, eliminating the need for an ethics committee as they are leftover biological samples collected by routine health surveillance services - Consultation with the Ethics Committee on the Use of Animals (CEUA nº 4434190521 / Federal University of Santa Catarina).”

Round 2

Reviewer 1 Report

Comments and Suggestions for Authors

No further comments.

Reviewer 2 Report

Comments and Suggestions for Authors

Dear Authors,

Thank you for corrected manuscript and replies. Your manuscript was improved.